# Copper-Decorated Ti$_3$C$_2$T$_x$ MXene Electrocatalyst for Hydrogen Evolution Reaction

**Buxiang Wang, Qing Shu \***, **Haodong Chen, Xuyao Xing, Qiong Wu and Li Zhang**

Faculty of Materials Metallurgy and Chemistry, Jiangxi University of Science and Technology, Ganzhou 341000, China
\* Correspondence: shuqing@jxust.edu.cn

**Abstract:** It remains a formidable challenge to prepare an economical and stable electrocatalyst for hydrogen evolution reaction using non-precious metals. In this study, MXene (Ti$_3$C$_2$T$_x$) nanosheets were prepared by high-energy ultrasound treatment, and Cu nanoparticles were prepared by NaBH$_4$ as a reducing agent. Then, the electrocatalyst Cu/Ti$_3$C$_2$T$_x$, suitable for hydrogen evolution reaction (HER), was prepared by supporting Cu with Ti$_3$C$_2$T$_x$. The structure, morphology, crystal phase and valence state of the obtained catalyst were determined by a variety of characterization analysis methods, and the influence of these properties on the catalytic performance is discussed here. The results of Brunner–Emmet–Teller (BET) showed that Ti$_3$C$_2$T$_x$ can effectively inhibit Cu agglomeration. Results of Transmission Electron Microscopy (TEM) and X-ray Diffraction (XRD) showed that Cu has metallic and oxidized states. X-ray Photoelectron Spectroscopy (XPS) further revealed the existence of multivalent states in Cu, which would contribute to the formation of electron transfer channels and the enhancement of electrocatalytic activity. In addition, the Cu/Ti$_3$C$_2$T$_x$ catalyst has strong hydrophilicity, as measured by contact angle, which is conducive to HER. Ti$_3$C$_2$T$_x$ has acceptable electrocatalytic hydrogen evolution performance: under alkaline conditions, when the current density is 10 mA cm$^{-2}$, HER overpotential is as low as 128 mV and the Tafel slope is as low as 126 mV dec$^{-1}$. Meanwhile, Ti$_3$C$_2$T$_x$ showed adequate stability for HER (94.0% of the initial mass activity after 1000 CV cycles). This work offers insights into the development of high-performance non-precious metal-based catalysts to achieve the high performance of HER in alkaline electrolytes.

**Keywords:** hydrogen evolution reaction; electrocatalyst; Cu; MXene (Ti$_3$C$_2$T$_x$) nanosheets





## 1. Introduction

Hydrogen energy, which is clean, efficient, safe and sustainable, has attracted extensive attention from researchers around the world in the face of global energy shortages and increasingly serious environmental pollution [1,2]. It has potential in the field of transportation and industrial production, so it is considered as one of the most promising energy types of the 21st century. At present, electrolysis of water is considered a mature and promising method for hydrogen production. However, the cathodic reaction (hydrogen evolution reaction, HER) and anodic reaction (oxygen evolution reaction, OER) have the problems of high energy consumption and instability in the process of hydrogen production from water electrolysis [3]. The theoretical thermodynamic voltage E$_0$ is 1.23 V when the electrolytic water reaction is carried out at a standard atmospheric pressure and 25 °C [4,5]. However, due to the influence of electrochemical resistance, transmission resistance, external circuit resistance and other factors, the voltage E applied at both ends of the electrode must be greater than 1.23 V. Therefore, the current technical research mainly focuses on how to reduce the energy loss and improve the energy conversion efficiency in the process of hydrogen production from water electrolysis.

Efficient electrocatalysts play an important role in the process of hydrogen production from water electrolysis, including reducing the reaction barrier, reducing energy consumption, increasing reaction efficiency and rate, etc. Therefore, the research of efficient

electrocatalysts has become the most important task in the exploration of electrolysis water hydrogen production technology. At present, the main electrocatalysts used in water electrolysis for hydrogen production are precious metals (Pt, Ru, Pd, Ir, etc.) and their alloys. Although such electrocatalysts have low overpotential, they also have the defects of high production cost and unscalable production, which cannot reach the standard of industrial scale of water electrolysis for hydrogen production [6,7]. Therefore, researchers have turned their attention to the relatively inexpensive non-precious metals (Ni, V, Mo, Fe, Cr, W, etc.) and their alloys [8]. At present, Cu-based catalysts have been found to be effective in electrocatalytic oxidation of glucose and reduction of $H_2O_2$ [9,10]. However, there is still a gap in the electrocatalytic performance of Cu-based catalysts when it is compared with nickel-based and cobalt-based catalysts. This limits the application of Cu-based catalysts in HER and OER [11]. At present, with the unremitting efforts of researchers, the catalytic performance of Cu-based catalysts in electrolysis of water for hydrogen production has been improved. Hyunsik et al. obtained two-dimensional copper oxide (CuO) nanosheets on stainless steel plates by chemical bath deposition and air annealing. They were found to have high catalytic activity and stability when used in OER [12].

Metal catalysts are usually prepared by gas-phase reduction, that is, by adding $H_2$ or CO under heating conditions to reduce the corresponding metal oxides. The reduction process will release a lot of heat, but the copper nanoparticles are extremely sintered ($\geq$300 °C) to render deactivation caused by the low Taman temperature of 407 °C, and overheating will lead to the condensation of the microcrystals of Cu, resulting in the reduction in catalytic activity [13,14]. Additionally, the severe clustering of nanoparticles has a negative impact on the catalytic performance [15]. In order to reduce the agglomeration of copper during preparation, Zhou et al. prepared Cu-based catalysts by using $NaBH_4$ as a reducing agent [16]. In contrast to the gas-phase reduction method, because this method is carried out in a solution at room temperature, the solvent in the solution can both disperse the reactants and rapidly absorb the heat generated by the exothermic reaction, thus slowing down Cu agglomeration during the preparation. In addition, some studies have found that suitable support not only provides a good support structure, but also can prevent the sintering of active substances in the process of use. It improves the heat resistance and prevents deactivation of the catalyst. Spencer found that Cu supported on ZnO can enhance the spillover effect of hydrogen atoms, adsorb water, disperse the active site of Cu, reduce the sintering rate of Cu particles and keep the small grains of copper in a metastable state [17]. MXenes are a diverse group of low-dimensional transition metal carbide and nitride with the general formula of $M_{n+1}X_nT_x$ (n = 1–3), where M is the transition metal; X is a carbide, nitride or carbonitride; and $T_x$ is the surface termination group. The surface of MXenes ($Ti_3C_2T_x$) is rich in O, OH and F groups, which provide abundant available sites for Cu adsorption [18]. $Cu^{2+}$ can be loaded onto the surface of $Ti_3C_2T_x$ nanosheets by electrostatic interaction [19]. Furthermore, Cu strongly reacts with oxygen-containing surface functional groups on the surface of $Ti_3C_2T_x$ nanosheets, which further promotes Cu adsorption and fixation on MXenes. In addition, Cu nanoparticles are inserted into the interspace between the MXene layers, effectively inhibiting the accumulation of MXenes and expanding the layer spacing, exposing more active sites, thereby improving its electrocatalytic performance [20,21].

In this study, MXene ($Ti_3C_2T_x$) nanosheets were prepared by high-energy ultrasound treatment, and Cu nanoparticles were prepared by $NaBH_4$ as a reducing agent. Then, the electrocatalyst $Cu/Ti_3C_2T_x$, suitable for HER, was prepared by supporting Cu with $Ti_3C_2T_x$. The structure, morphology, crystalline phase and valence states of the samples were analyzed by several characterization methods. The activity and stability of $Cu/Ti_3C_2T_x$ for HER were studied. The reasons for the improvement of electrocatalytic performance are discussed here, such as the influence of the presence of multiple electron valence states in copper on electron transport, $Ti_3C_2T_x$ on Cu agglomeration, and strong hydrophilicity of $Ti_3C_2T_x$ catalyst on water splitting.

## 2. Materials

All reagents were of analytic reagent grade and used as received without additional purification. Copper nitrate trihydrate ($Cu(NO_3)_2 \cdot 3H_2O$), titanium aluminum carbide 312($Ti_3AlC_2$), lithium fluoride (LiF) and sodium borohydride ($NaBH_4$) were purchased from Aladdin Corporation (Ontario, CA, USA).

### 2.1. Preparation of $Cu/Ti_3C_2T_x$ Catalyst

$Ti_3C_2T_x$: First, we put 1 g lithium fluoride (LiF) into a Teflon beaker containing 20 mL hydrochloric acid (HCl 12 mol/L). Then, we put the beaker containing the mixture on the constant-temperature magnetic stirrer and set the temperature to 55 °C and the rate of magnetic stirring to 100 rpm. After LiF was completely dissolved in hydrochloric acid, 1 g of titanium aluminum carbide ($Ti_3AlC_2$) powder was slowly placed into a beaker containing the mixture. The mixture was stirred continuously for 24 h to remove the Al metal layer. Second, the obtained suspension was decanted into a centrifuge tube and centrifuged at 6400 rpm for 10 min to stratify the suspension. The upper layer was clear liquid and the lower layer was sediment. We poured the supernatant out of the centrifuge tube, added a certain amount of deionized water and shook well, and then continued to centrifuge at the rate of 6400 rpm for 10 min, so that the suspension was stratified again. After that, we poured out the liquid on the top of the centrifuge tube, refilled it with deionized water and shook well. This process was repeated until the pH of the mixture reached 6 to ensure that residual HF and other impurities were removed. Finally, the suspension was filtered by using a suction filter, and the collected sediment was dried in an oven at 60 °C for 24 h to obtain $Ti_3C_2T_x$ ($T_x$ stands for O, F, OH functional group). The illustration of the $Cu/Ti_3C_2T_x$ synthetic is shown in Figure 1.

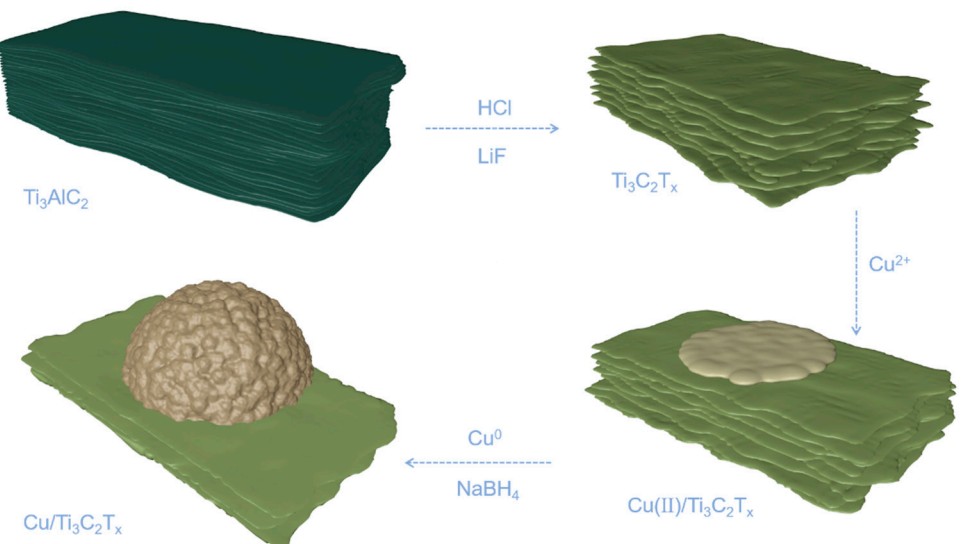

**Figure 1.** The illustration of the synthetic for $Cu/Ti_3C_2T_x$.

$Cu/Ti_3C_2T_x$: In order to synthesize $Ti_3C_2T_x$ nanosheets, 0.3 g $Ti_3C_2T_x$ was poured into a beaker containing 30 mL deionized water and sonicated in an ice water bath for 3 h. We placed the beaker filled with $Ti_3C_2T_x$ on the thermostatic magnetic stirrer, set the temperature to 25 °C, added 0.6 g of $Cu(NO_3)_2$ to the $Ti_3C_2T_x$ solution, stirred at 800 RPM for 30 min and let stand for another 3 h until the solution was clear and layered (the top solution was light blue). We added a certain amount of $NaBH_4$ until the solution was clear and layered (the upper layer was colorless and transparent), stirred for 30 min at 800 rpm and let stand for 12 h. The upper layer of the supernatant in the beaker was poured out and the remaining pellet was dried in a freeze dryer for 48 h to obtain $Cu/Ti_3C_2T_x$.

To expand the comparison scope, the preparation of Cu and $Ti_3C_2T_x$ was similar to the above. $Cu(NO_3)_2$ solution was directly added to $NaBH_4$ to obtain Cu, and $Ti_3C_2T_x$ solution after ultrasonic dispersion was directly dried to obtain $Ti_3C_2T_x$.

### 2.2. Material Characterization

The crystalline phases of $Ti_3AlC_2$, $Ti_3C_2T_x$ and $Cu/Ti_3C_2T_x$ were analyzed by using XRD (Empyrean, Panalytical, Almelo, The Netherlands). The microstructure of $Ti_3C_2T_x$ and $Cu/Ti_3C_2T_x$ catalysts was characterized by SEM (MLA650F, FEI, Hillsboro, OR, USA) and TEM (Tecnai G2-20, FEI, Hillsboro, OR, USA). In addition, the elemental phases of the $Cu/Ti_3C_2T_x$ catalyst were analyzed by Scanning Electron Microscopy coupled with Energy-Dispersive X-ray Spectroscopy (SEM-EDX, MLA650F, FEI, Hillsboro, OR, USA). Surface chemical states of the $Cu/Ti_3C_2T_x$ catalyst were determined by XPS (ESCALAB Xi+, Waltham, MA, USA). The surface wetting performance of $Ti_3C_2T_x$, Cu and $Cu/Ti_3C_2T_x$ was tested on the contact angle meter (JCY type, ShangHai Fangrui Instrument Co., Ltd., Shanghai, China). The specific surface areas of $Ti_3C_2T_x$, Cu and $Cu/Ti_3C_2T_x$ were determined by a $N_2$ physical adsorption instrument (ASAP2020, Micromeritics Instrument Corporation, Austin, TX, USA).

### 2.3. Electrochemical Measurements

The electrochemical properties of Cu, $Ti_3C_2T_x$ and $Cu/Ti_3C_2T_x$ were tested by using the electrochemical workstation (CHI660E). A carbon rod acted as the counter electrode (CE), a clip electrode coated with the sample material acted as the working electrode (WE) and a calomel electrode acted as the auxiliary electrode (AE). The catalyst was placed in a mixture of isopropyl alcohol and water solvent, and ultrasonic action was performed for 40 min. Then, it was coated on the carbon paper of the clip electrode and dried at 25 °C. All tests were conducted in a 1 M KOH solution and the test ambient temperature was controlled at 25 °C. All potential values were converted to standard electrode potential relative to the reversible hydrogen electrode (RHE), calculated as follows:

$$V_{RHE} = V_{(SCE)} + 0.0591 \times pH + 0.2415 \tag{1}$$

HER activities were measured by linear sweep voltammetry (LSV). The overpotential ($\eta$) of the cathode can be obtained from the LSV graph. In this work, the scanning rate in the LSV test was 5 mV s$^{-1}$. In addition, the slope of the Tafel plot can also be calculated from the LSV graph. To quantitatively determine the double-layer capacitance (*C*dl) of the catalyst, the current density of the catalyst was tested by using cyclic voltammetry (CV) and setting different sweep rates. Since the value of the electrochemical specific surface area (ECSA) is proportional to the value of *C*dl, the ECSA of catalyst can be obtained from the *C*dl value. To test the stability of the catalyst, long-term cyclic voltammetry was used.

### 3. Results and Discussions

#### 3.1. Characterization of Cu/$Ti_3C_2T_x$ Catalyst

##### 3.1.1. SEM

The SEM images clarify the morphology and microstructures of $Cu/Ti_3C_2T_x$. $Ti_3C_2T_x$ exhibits a distinct layered structure [22]. After HF etches, various surface terminations are formed on the surface of $Ti_3C_2T_x$ [23], which can provide abundant sites for Cu atoms anchored on the $Ti_3C_2T_x$. $Cu^{2+}$ can be reacted with surface terminations of $Ti_3C_2T_x$, which transforms into $Cu^{1+}$ through self-induced redox reactions [24]. Figure 2a is the SEM image of $Cu/Ti_3C_2T_x$. These particles loaded on the surface of $Ti_3C_2T_x$ are Cu, which can be inserted into the gaps between the layers [25]. In addition, Figure 2b displays the elemental distribution maps of $Cu/Ti_3C_2T_x$, showing the distribution of elements in $Cu/Ti_3C_2T_x$ by SEM-EDX, and indicating the presence of C, O, Ti and Cu distributed uniformly throughout the catalyst surface. In addition, the estimated atomic contents of C, O, Ti and Cu are about

26.98%, 33.72%, 9.70% and 29.59%, respectively. The atomic proportion of O is the highest in the $Cu/Ti_3C_2T_x$, indicating a potential presence of oxidized copper.

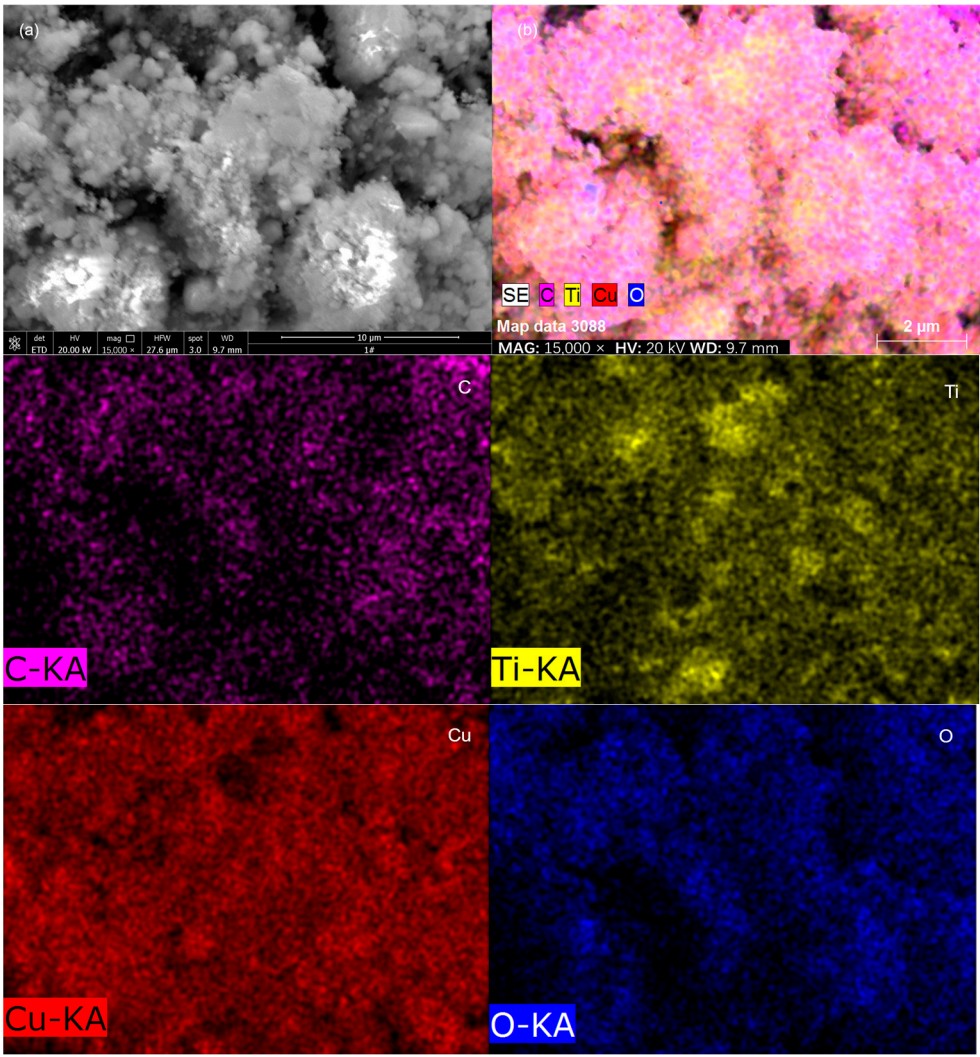

**Figure 2.** SEM image of (**a**) $Cu/Ti_3C_2T_x$ and (**b**) EDS mapping of $Cu/Ti_3C_2T_x$.

### 3.1.2. TEM

In order to obtain more detailed structural information of the catalyst, the synthesized $Ti_3C_2T_x$ and $Cu/Ti_3C_2T_x$ nanostructures were investigated by TEM. Figure 3a shows a typical sheet-like morphology of $Ti_3C_2T_x$. Figure 3b shows structural characteristics of $Cu/Ti_3C_2T_x$. Numerous particles scatter across $Ti_3C_2T_x$. According to the analysis of the SEM results, $Cu^{2+}$ in solution is reduced to $Cu^0$ by $NaBH_4$, which aggregates into small particles and covers the surface of $Ti_3C_2T_x$ [25]. Moreover, gaps and long cones between these particles can be observed, which are formed by a large number of reduced $Cu^0$ and $Cu^{1+}$ ions that grow in a certain direction [26]. As can be seen from Figure 3c, the interplanar spacing of 0.21 nm can be assigned to the (111) crystallographic plane of Cu phase [27,28]. In Figure 3d, the interplanar spacing of 0.25 nm corresponds to the (111) plane of $Cu_{2+1}O$ phase [29], indicating the presence of metallic and oxidized states in $Cu/Ti_3C_2T_x$.

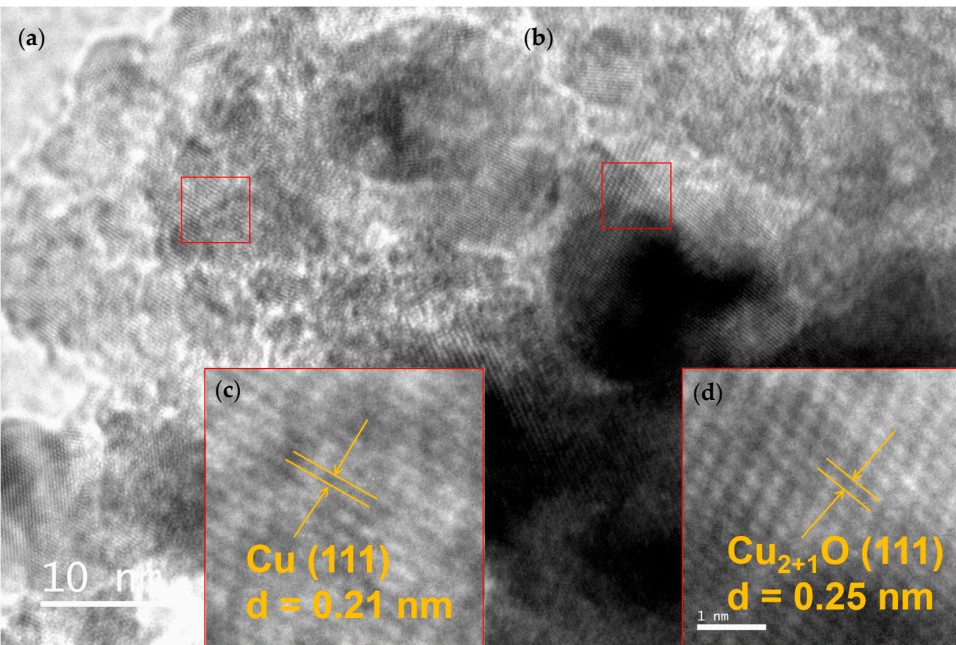

**Figure 3.** TEM image of (**a**) Ti$_3$C$_2$T$_x$, (**b–d**) Cu/Ti$_3$C$_2$T$_x$.

### 3.1.3. Physical Adsorption/Desorption Measurements of N$_2$

The specific surface areas of Ti$_3$C$_2$T$_x$, Cu and Cu/Ti$_3$C$_2$T$_x$ were determined by the Brunner–Emmet–Teller (BET) method, and the test was performed on a N$_2$ physical adsorption instrument. N$_2$ adsorption isotherms of Ti$_3$C$_2$T$_x$, Cu and Cu/Ti$_3$C$_2$T$_x$ are given in Figure 4. The specific surface area of Ti$_3$C$_2$T$_x$ is 2.9 m$^2$ g$^{-1}$. Ti$_3$C$_2$T$_x$ is easy to stack together, thus losing the characteristic of a high specific surface area [30]. The specific surface area of Cu is 6.4 m$^2$ g$^{-1}$, which can be attributed to the easy agglomeration of nanoscale Cu$^0$ [31]. The specific surface area of the Cu/Ti$_3$C$_2$T$_x$ is 9.5 m$^2$ g$^{-1}$, which indicates the successful preparation of a Cu/Ti$_3$C$_2$T$_x$ catalyst with a high specific surface area. This can be attributed to the fact that Cu nanoparticles effectively inhibit the accumulation of Ti$_3$C$_2$T$_x$ [20,21]. It is worth noting that the number of active sites on the surface of a catalyst is positively correlated with its catalytic performance. Catalysts with large specific surface areas can provide more active sites for HER, helping to reduce the overpotential of electrochemical reactions [32]. Put another way, the actual surface of the catalyst is larger than the macroscopic geometric area, which can make it have more active sites and shorten the ion migration distance, thus facilitating the release of H$_2$ in alkaline solution.

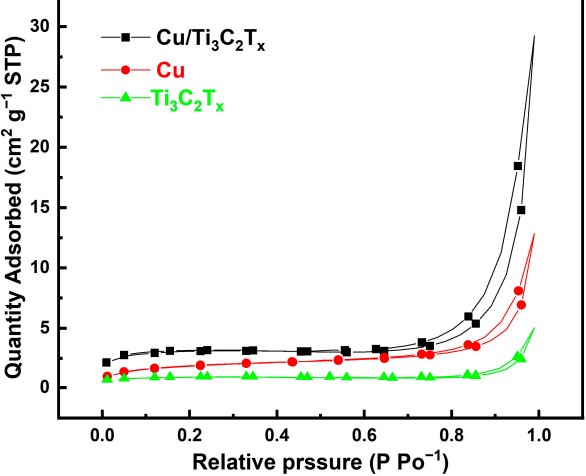

**Figure 4.** N$_2$ adsorption/desorption isotherms for Ti$_3$C$_2$T$_x$, Cu and Cu/Ti$_3$C$_2$T$_x$.

### 3.1.4. XRD

The crystal structure of $Ti_3AlC_2$, $Ti_3C_2T_x$ and $Cu/Ti_3C_2T_x$ can be reflected by the XRD diffraction pattern. As shown in Figure 5a, $Ti_3AlC_2$ has pronounced characteristic diffraction peaks at 2θ of 9.5°, 19.1° and 39.0°, which are consistent with the data of $Ti_3AlC_2$ (JCPDS. NO.52-0875), corresponding to $Ti_3AlC_2$ (002), (004) and (104) crystal planes, respectively. After etching, diffraction peaks at 9.5°, 19.1° and 39.0° (2θ) are all significantly attenuated. Moreover, a new peak appears at 6.1°, indicating that $Ti_3AlC_2$ has successfully transitioned into $Ti_3C_2T_x$ [33,34]. The XRD pattern of $Ti_3C_2T_x$ is shown in Figure 5b, and 10 characteristic diffraction peaks appear at 2θ of 34.0°, 36.6°, 39.1°, 41.6°, 48.3°, 56.7°, 60.0°, 65.3°, 70.2° and 73.9°, corresponding to $Ti_3AlC_2$ (101), (103), (104), (105), (107), (109), (110), (1011), (1012) and (108) crystal planes, respectively. At the same time, three characteristic diffraction peaks appeared at 2θ of 35.9°, 41.7° and 60.4°, basically consistent with TiC (JCPDS. NO.32-1383), corresponding to TiC (111), (200) and (220) crystal planes, respectively, which indicates that the inevitable presence of TiC and $Ti_3AlC_2$ impurities in the product is illustrated after etching [35]. In Figure 5c, four peaks appear at 2θ of 43.2°, 50.4°, 74.1° and 89.9°, respectively, and these characteristic peaks are consistent with the standard of Cu (JCPDS. NO.04-0836), corresponding to the (111), (200), (220) and (311) crystal planes of Cu, respectively. In addition, $Cu/Ti_3C_2T_x$ also appeared at 2θ of 29.5°, 36.4°, 42.2°, 52.4°, 61.3°, 69.5°, 73.5° and 77.3°, and these peaks are related to $Cu_{2+1}O$ (JCPDS.NO.05-0667), corresponding to $Cu_{2+1}O$ (110), (111), (200), (211), (220), (310), (311) and (222) crystal planes, respectively. Additionally, a peak appears at 60.7° related to $Ti_3C_2T_x$ [34], which suggests that $Cu/Ti_3C_2T_x$ has been successfully synthesized. In addition, the analysis result of XRD is consistent with the results of the TEM, which further confirms the presence of metallic (Cu) and oxidized ($Cu_{2+1}O$) states in $Cu/Ti_3C_2T_x$.

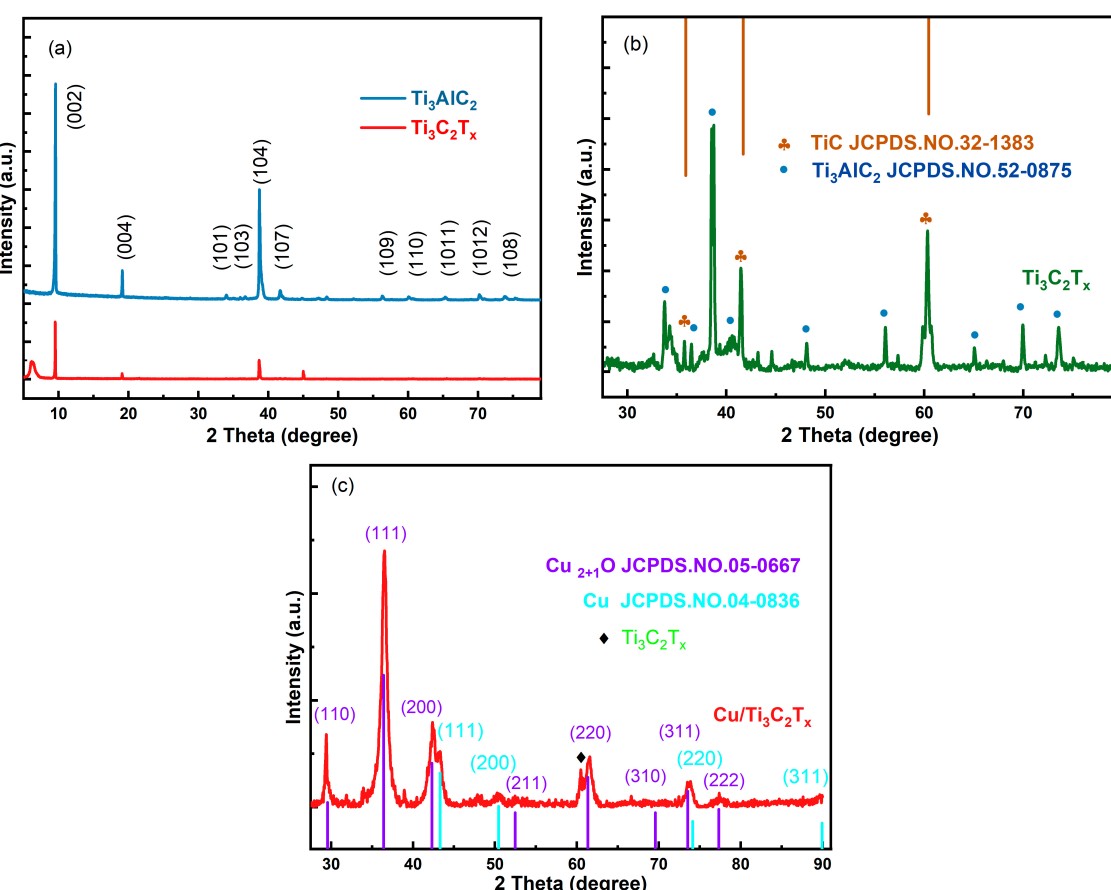

**Figure 5.** XRD pattern of (**a**) $Ti_3AlC_2$ and $Ti_3C_2T_x$; XRD pattern of (**b**) $Ti_3C_2T_x$ and (**c**) $Cu/Ti_3C_2T_x$.

### 3.1.5. XPS

The surface chemical composition of the $Cu/Ti_3C_2T_x$ catalyst was identified by XPS, which can further investigate the oxidation state and composition of $Cu/Ti_3C_2T_x$. In Figure 6a, the XPS survey scan band of the $Cu/Ti_3C_2T_x$ catalyst reveals the occurrence of Cu, C, O and Ti elements in the catalyst. Figure 6b shows the XPS Cu 2p spectrum of the $Cu/Ti_3C_2T_x$ catalyst; peaks at 935.1 eV ($Cu\ 2p_{3/2}$) and 954.7 eV ($Cu\ 2\ p_{1/2}$) can be attributed to $Cu^{2+}$ [36]. Peaks at $Cu\ 2p_{1/2}(952.0)$ and $Cu\ 2p_{3/2}(932.2\ eV)$ correspond to $Cu^+$ [37]. In addition, peaks at 932.1 eV ($Cu\ 2p_{3/2}$) and 952.5 eV ($Cu\ 2p_{1/2}$) can be attributed to $Cu^0$ [27]. This indicates the presence of Cu and $Cu_{2+1}O$ in $Cu/Ti_3C_2T_x$ materials [38]. The C1s spectrum of the $Cu/Ti_3C_2T_x$ catalyst is shown in Figure 6c, containing C-Ti (281.8 eV), C-C (284.8 eV), O-C (286.4 eV) and O-C=O (289.0 eV) bonds [39,40]. Figure 6d shows the O 1s spectrum of the $Cu/Ti_3C_2T_x$ catalyst. The peak at 532.9 eV conforms to H-O-H bonds [41], which are adsorbed on the surface of $Cu_{2+1}O$-doped Cu [42,43]. The peak at 531.7 eV corresponds to O-H, while the peak at 530.5 eV may correspond to $Cu_{2+1}O$. This shows that the surface of the catalyst contains not only Cu oxides (Cu-O), but also hydroxides (Cu-OH), and Cu oxide/hydroxide catalysts can effectively accelerate the dissociation of water [44–46]. Moreover, CuO is a hydrophilic material [47,48], which also verifies that the $Cu/Ti_3C_2T_x$ catalyst has a certain hydrophilicity.

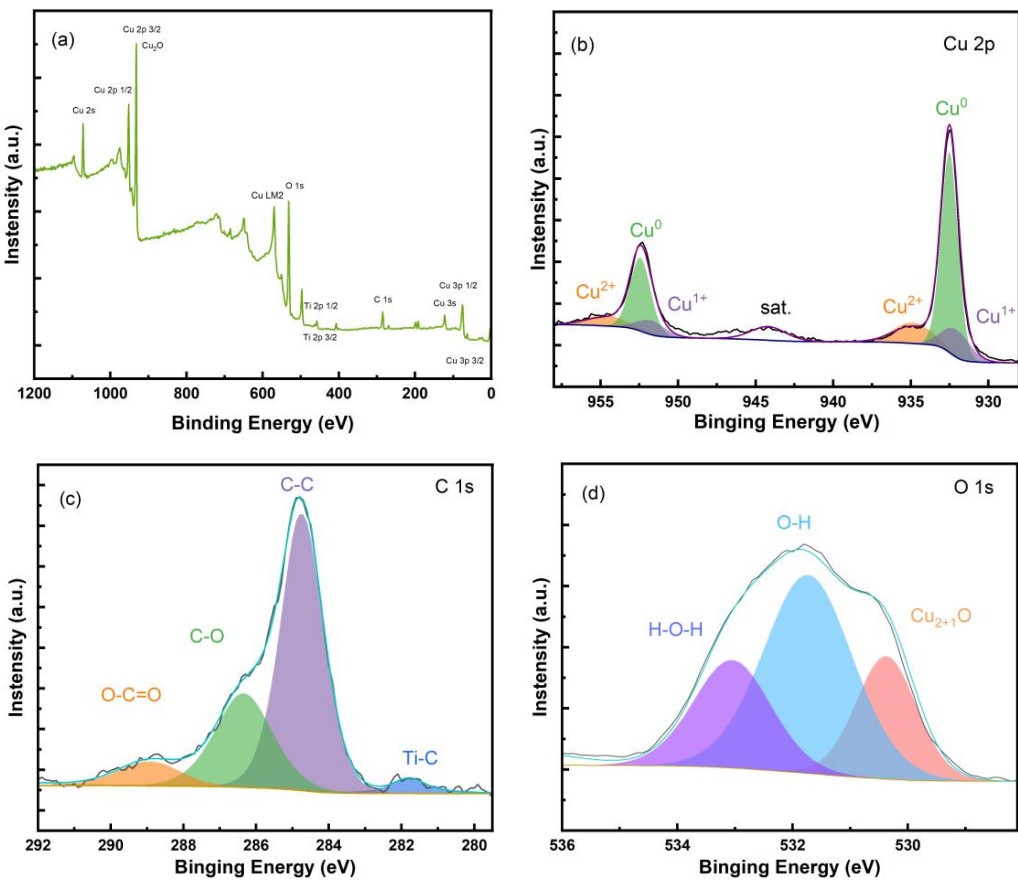

**Figure 6.** XPS spectra of $Cu/Ti_3C_2T_x$: (**a**) full spectrum of $Cu/Ti_3C_2T_x$, (**b**) Cu 2p spectra of $Cu/Ti_3C_2T_x$, (**c**) C 1s spectra of $Cu/Ti_3C_2T_x$, (**d**) O 1s spectra of $Cu/Ti_3C_2T_x$.

### 3.1.6. Contact Angle Test

The contact angle is a quantitative measure of the wettability of the surface. To further test the hydrophilicity of $Ti_3C_2T_x$, Cu and $Cu/Ti_3C_2T_x$ catalysts, the contact angle between the drop and the catalyst surface was measured. In Figure 7a, the contact angle of $Ti_3C_2T_x$ is 142.3°. In Figure 7b, the contact angle of Cu is 33.1°. In Figure 7c, the contact angle of

Cu/Ti$_3$C$_2$T$_x$ is 12.7°. The contact angle of Cu/Ti$_3$C$_2$T$_x$ is much smaller than that of Ti$_3$C$_2$T$_x$ and Cu, which indicates that the Cu/Ti$_3$C$_2$T$_x$ catalyst has stronger hydrophilicity. In light of the SEM and TEM analysis results, Cu nanoparticles are inserted into the interspace between the Ti$_3$C$_2$T$_x$ layers, effectively inhibiting the accumulation of Ti$_3$C$_2$T$_x$, exposing more hydrophilic functional groups, thereby improving its hydrophilicity. In the hydrogen evolution reaction, the stronger the hydrophilicity of the catalyst, the shorter the time that the bubbles generated by the reaction stay on the catalyst surface, and the smaller the size of the bubbles leaving the surface of the catalyst, which is more conducive to the improvement of the efficiency of the electrolysis water reaction [49].

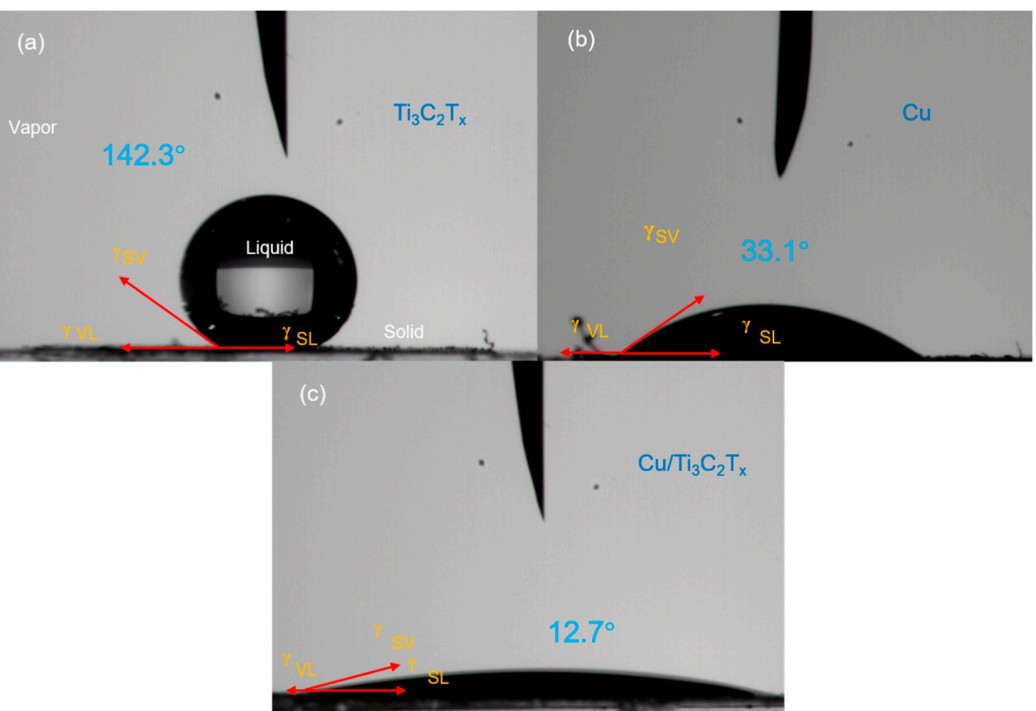

**Figure 7.** The contact angle test of Ti$_3$C$_2$T$_x$, Cu and Cu/Ti$_3$C$_2$T$_x$; (**a**) Ti$_3$C$_2$T$_x$, (**b**) Cu and (**c**) Cu/Ti$_3$C$_2$T$_x$.

### 3.2. HER Activity in Alkaline Electrolytes

The electrocatalytic properties of the synthesized Cu/Ti$_3$C$_2$T$_x$ electrocatalysts were investigated by the CV measurement at different sweep speeds. As shown in Figure 8a, the area of the closed curve increases with increasing scanning speed. To expand the comparison scope, the $C_{dl}$ of the catalysts was defined by the CV of the synthesized Cu, Ti$_3$C$_2$T$_x$ and Cu/Ti$_3$C$_2$T$_x$. As shown in Figure 8b, the $C_{dl}$ of the Cu/Ti$_3$C$_2$T$_x$ is 29.4 mF cm$^{-2}$. The $C_{dl}$ values of the Ti$_3$C$_2$T$_x$ and Cu are 9.8 and 2.1 mF cm$^{-2}$, respectively. The $C_{dl}$ of Cu/Ti$_3$C$_2$T$_x$ is 3 times higher than that of Ti$_3$C$_2$T$_x$. Furthermore, the ECSA of the catalysts can be obtained from the $C_{dl}$ value by a formulated methodology [50]. As shown in Figure 8c, the ECSA of Cu/Ti$_3$C$_2$T$_x$ is 735 cm$^{-2}$. The ECSA values of Ti$_3$C$_2$T$_x$ and Cu are 245 and 52.5 cm$^{-2}$, respectively. The ECSA of Cu/Ti$_3$C$_2$T$_x$ is 3 times higher than that of Ti$_3$C$_2$T$_x$. The large ECSA value can usually be attributed to the unique structure of Cu/Ti$_3$C$_2$T$_x$. Ti$_3$C$_2$T$_x$ supports small Cu nanoparticles, which not only effectively prevents Cu agglomeration, but also effectively improves the specific surface area of Cu/Ti$_3$C$_2$T$_x$ catalyst.

The HER performance of Ti$_3$C$_2$T$_x$, Cu and Cu/Ti$_3$C$_2$T$_x$ was investigated in 1 M KOH solution. In Figure 9a, Ti$_3$C$_2$T$_x$ exhibits the worst HER activity and Cu/Ti$_3$C$_2$T$_x$ exhibits the best HER performance. The catalytic activity exhibits the following trend: Cu/Ti$_3$C$_2$T$_x$ > Cu > Ti$_3$C$_2$T$_x$. The improvement of the catalytic performance of Cu/Ti$_3$C$_2$T$_x$ can be attributed to the insertion of Cu nanoparticles into Ti$_3$C$_2$T$_x$, which improves the catalytic

activity and enhances the conductivity of $Ti_3C_2T_x$-based catalysts [51]. $Cu/Ti_3C_2T_x$ shows the lowest overpotential ($\eta_{10}$ = 128 mV), while that of Cu is 360 mV and that of $Ti_3C_2T_x$ is 541 mV. In addition, the overpotential ($\eta_{10}$) of $Cu/Ti_3C_2T_x$ is lower than that of most of the non-precious catalysts for HER listed in Table 1 [52–59].

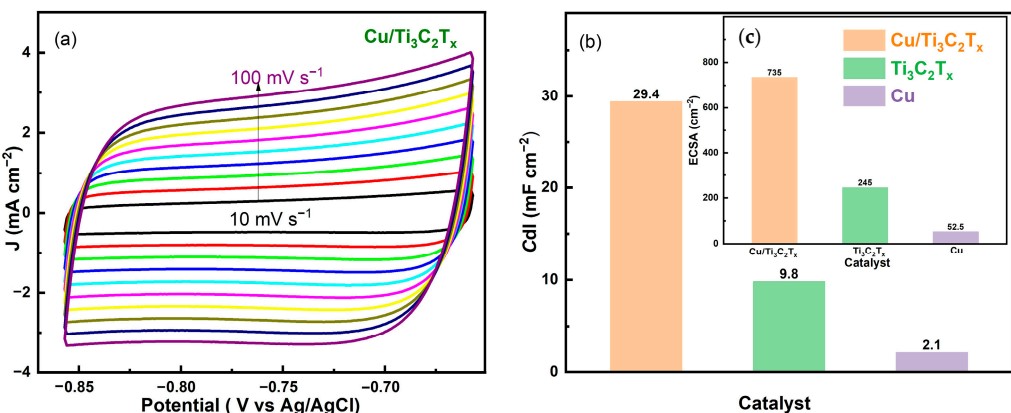

**Figure 8.** HER performance of $Ti_3C_2T_x$, Cu and $Cu/Ti_3C_2T_x$ series catalysts: (**a**) CV curves of $Cu/Ti_3C_2T_x$ at the scan rate of 10, 20, 30, 40, 50, 60, 70, 80, 90, 100 mV s$^{-1}$, (**b**) comparisons of Cdl, (**c**) comparisons of ECSA.

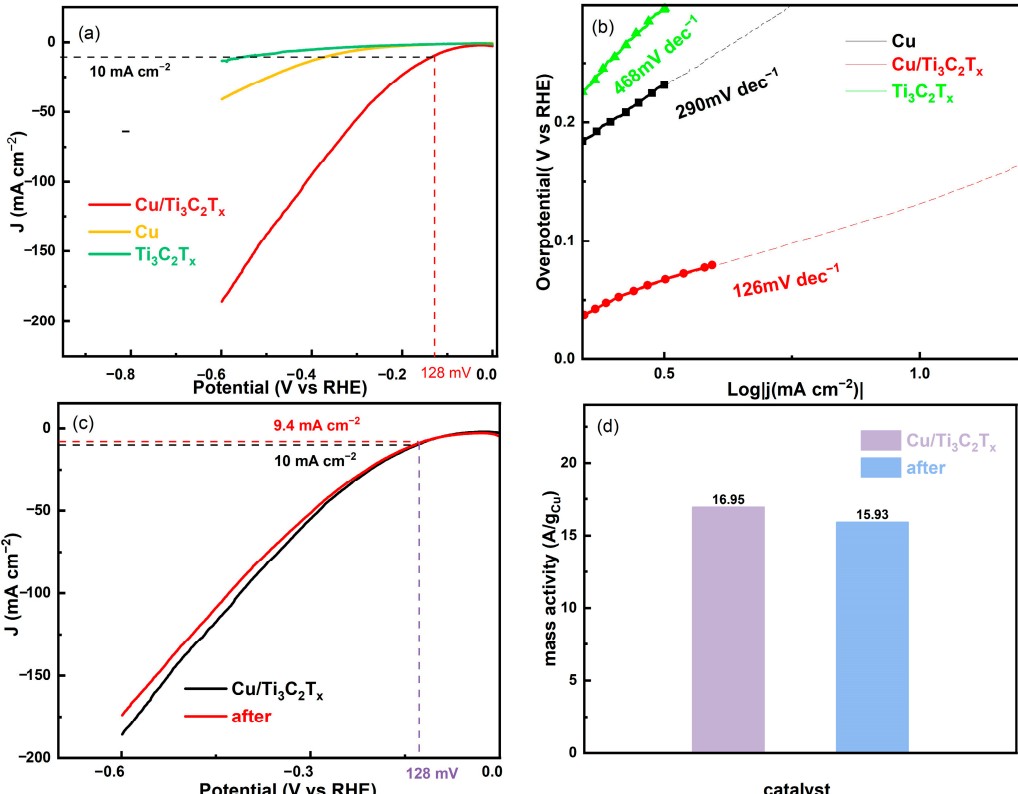

**Figure 9.** HER performance of $Ti_3C_2T_x$, Cu and $Cu/Ti_3C_2T_x$ series catalysts: (**a**) LSV curves at the scan rate of 5mV s$^{-1}$, (**b**) the corresponding Tafel plots, (**c**) stability test chart of $Cu/Ti_3C_2T_x$, (**d**) comparisons of mass activity of $Cu/Ti_3C_2T_x$.

The Tafel slope is an important parameter in evaluating the mechanistic pathway of the hydrogen evolution reaction. The lower the value of Tafel, the stronger the HER catalytic activity of the catalyst. In Figure 9b, the Tafel slope ($Cu/Ti_3C_2T_x$) is 126 mV dec$^{-1}$, which is less than that of $Ti_3C_2T_x$ (468 mV dec$^{-1}$) and Cu (290 mV dec$^{-1}$), indicating that the dynamics for HER of $Cu/Ti_3C_2T_x$ are faster [60]. In addition, the value of the Tafel

slope infers the rate controlling step of the reaction, and the Tafel slope of the $Cu/Ti_3C_2T_x$ catalyst is 126 mV $dec^{-1}$, corresponding to the Volmer–Heyrovsky mechanism, while the rate controlling step of this reaction is a Volmer step [61,62].

**Table 1.** The overpotential of non-precious metal-based HER catalysts.

| Catalyst | Overpotential ($\eta_{10}$) | References |
|:---:|:---:|:---:|
| $Cu/Ti_3C_2T_x$ | 128 mV | This work |
| $Mo-Ti_2Cu_3$ | 133 mV | [52] |
| $Cu_2(OH)PO_4/Co_3(PO_4)_2\cdot 8H_2O$ | 138 mV | [53] |
| $Cu_2O/g-C_3N_4$ | 148.7 mV | [54] |
| PEDOT@Mn-Salen COFEDA | 150 mV | [55] |
| $Cu_3P$ | 155 mV | [56] |
| fs-$Cu/MoS_2$ | 181 mV | [57] |
| $Cu(OH)_2$@FCN MOF/CF | 290 mV | [58] |
| $Cu_2S$ | 330 mV | [59] |

It is worth noting that the HER under alkaline conditions involves the dissociation of water molecules (Volmer reaction), and water molecules must first adsorb and dissociate on the surface of the catalyst to form adsorption states of $H_{ads}$ and $OH^-$ [63]. In light of the SEM, TEM, XRD and XPS analysis results, CuO and $Cu(OH)_2$ contained in $Cu/Ti_3C_2T_x$ promote the water dissociation and improve the overall efficiency of water electrolysis. Figure 9c depicts the LSV curve after 1000 CV cycles under alkaline conditions with a current density of 9.4 mA $cm^{-2}$ at an overpotential of 128 mV. Compared with the current density of $Cu/Ti_3C_2T_x$ before the test, the current density of $Cu/Ti_3C_2T_x$ at an overpotential of 128 mV after the test only decreases by 0.6 mA $cm^{-2}$. As shown in Figure 9d, the mass activity of $Cu/Ti_3C_2T_x$ at an overpotential of 128 mV is 16.95 A/gCu. After the test, the mass activity of $Cu/Ti_3C_2T_x$ decreases by only 1.02 A/gCu, indicating that $Cu/Ti_3C_2T_x$ has better stability.

## 4. Conclusions

An electrocatalyst for hydrogen evolution, $Cu/Ti_3C_2T_x$, was synthesized by loading Cu onto $Ti_3C_2T_x$. $Cu/Ti_3C_2T_x$ shows efficient electrocatalytic performance. Under alkaline conditions, the HER overpotential is as low as 128 mV at a current density of 10 mA $cm^{-2}$. The Tafel slope of the $Cu/Ti_3C_2T_x$ catalyst is as low as 126 mV $dec^{-1}$. In addition, the mass activity of $Cu/Ti_3C_2T_x$ at an overpotential of 128 mV decreases by only 1.02 A/gCu after the stability test, which indicates that the $Cu/Ti_3C_2T_x$ has acceptable stability. The high catalytic performance of $Cu/Ti_3C_2T_x$ can be attributed to the following aspects: CuO and $Cu(OH)_2$ accelerate water dissociation, and the presence of CuO makes the $Cu/Ti_3C_2T_x$ catalyst more hydrophilic, which accelerates the release of bubbles and effectively improves the efficiency of the electrolyzed water reaction. This work provides insights into the development of high-performance non-precious metal-based catalysts to achieve the high performance of HER in alkaline electrolytes.

**Author Contributions:** Conceptualization, B.W. and Q.S.; methodology, Q.S.; software, Q.W. and L.Z.; validation, B.W.; formal analysis, B.W.; investigation, B.W.; resources, H.C. and X.X.; data curation, B.W.; writing—original draft, B.W.; writing—review and editing, Q.S. and B.W.; visualization, B.W.; supervision, Q.S.; project administration, B.W.; funding acquisition, Q.S. All authors have read and agreed to the published version of the manuscript.

**Funding:** This research was funded by [National Natural Science Foundation of China] grant number [21766009].

**Data Availability Statement:** The data presented in this study are available on request from the corresponding author. The data are not publicly available due to ethical.

**Conflicts of Interest:** The authors declare no conflict of interest. The funders had no role in the design of the study; in the collection, analyses, or interpretation of data; in the writing of the manuscript; or in the decision to publish the results.

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
