# Peer review of "Copper-Decorated Ti3C2Tx MXene Electrocatalyst for Hydrogen Evolution Reaction"

_metals, doi:10.3390/met12122022_

Round 1
Reviewer 1 Report
The reviewed manuscript entitled: “Copper decorated MXenes(Ti3C2Tx) electrocatalyst for high-ef-2 ficiency hydrogen evolution reaction” deals with design and evaluation of low HER overpotential Cu – based catalyst deposited onto layered MXene substrate by performing efficient chemical reduction method with the use of NaBH4 as reducing agent in mild conditions. Very prominent hydrophilization of the catalytic surface was observed after Cu clusters deposition which is important factor in suppression of unwanted surface bubble formation phenomena during electrolysis cell operation. The manuscript is well structured and well-written in terms of high scientific rigor and good presentation of the results achieved. I have only few critical remarks as follow:
Results and discussion
XPS: “Moreover, CuO 250 is a hydrophilic material [45,46], which also verifies that the Cu/Ti3C2Tx catalyst has a certain hydrophilia.” The authors used the term “hydrophilia” instead of hydrophilicity and if this term is really correct they have to give some more detailed explanation, otherwise pls use hydrophilicity or something like that instead. Also, lease critically read and may be include in the References (Introduction and/or Results and discussion part) the following published paper by Shahzad et al. “Two-Dimensional Ti3C2Tx MXene Nanosheets for Efficient Copper Removal from Water” ACS Sustainable Chemistry & Engineering 5(12); where the authors claim some hydrophilic properties of MXene itself which is in some contradiction with the current presented results or may be it depends on the water droplet time of contact and the kind of MXylene surface?
“The contact angle of Cu/Ti3C2Tx is much smaller than that of Ti3C2Tx and Cu, which indicates that the Cu/Ti3C2Tx catalyst has strong hydrophilia. In the hydrogen evolution reaction, the stronger the hydrophilia of the catalyst, the shorter the time that the bubbles generated by the reaction stay on the catalyst surface, and the smaller the size of the bubbles leaving the surface of the catalyst, which is more conducive to the improvement of the efficiency of the electrolysis water reaction.” Here again the term “hydrophilia”.
Author Response
Point-by-Point Response to Reviewers’ Comments
For your reference, the detailed point-by-point response to comments from referees is presented below. To save your and the reviewers’ time to quickly read what we have done for the revision, in this response letter, the texts for “Author reply” are marked in blue font, and the texts for “Corresponding revisions” are marked in red font. The revised or added texts for changes have been documented in red font in our revised manuscript.
Thank you for your attention in your busy time!
Response to the comments of referee 1
Comments: The reviewed manuscript entitled: “Copper decorated MXenes(Ti3C2Tx) electrocatalyst for high-efficiency hydrogen evolution reaction” deals with design and evaluation of low HER overpotential Cu-based catalyst deposited onto layered MXene substrate by performing efficient chemical reduction method with the use of NaBH4 as reducing agent in mild conditions. Very prominent hydrophilization of the catalytic surface was observed after Cu clusters deposition which is important factor in suppression of unwanted surface bubble formation phenomena during electrolysis cell operation. The manuscript is well structured and well-written in terms of high scientific rigor and good presentation of the results achieved. I have only few critical remarks as follow:
Author reply:
We deeply appreciate the reviewer’s agreement on this work. Your concerns have been carefully considered and we have revised our manuscript with great attention.
Comments 1: XPS: “Moreover, CuO 250 is a hydrophilic material [45,46], which also verifies that the Cu/Ti3C2Tx catalyst has a certain hydrophilia.” The authors used the term “hydrophilia” instead of hydrophilicity and if this term is really correct they have to give some more detailed explanation, otherwise pls use hydrophilicity or something like that instead. Also, lease critically read and may be include in the References (Introduction and/or Results and discussion part) the following published paper by Shahzad et al. “Two-Dimensional Ti3C2Tx MXene Nanosheets for Efficient Copper Removal from Water” ACS Sustainable Chemistry & Engineering 5(12); where the authors claim some hydrophilic properties of MXene itself which is in some contradiction with the current presented results or may be it depends on the water droplet time of contact and the kind of MXylene surface?
Author reply:
We deeply appreciate the reviewer’s agreement on this work. Your concerns have been carefully considered and we have revised our manuscript with great attention.
Mxene has high hydrophilicity due to the existence of hydrophilic functional groups (such as -F, -OH and O) on its surface. However, Ti3C2Tx solution after ultrasonic dispersion was directly dried to obtain Ti3C2Tx, which easily led to the occurrence of defects such as aggregation and collapse of the structure. In addition, Cu nanoparticles are inserted into the interspace between the MXene layers, effectively inhibiting the accumulation of MXenes and expanding the layer spacing, exposing more hydrophilic functional groups, thereby improving its electrocatalytic performance
According to your suggestion, this paragraph has been rewritten.
Corresponding revisions:
Page 9 of manuscript, Lines 283-285
In the light of the SEM and TEM analysis results, Cu nanoparticles are inserted into the interspace between the Ti3C2Tx layers, effectively inhibiting the accumulation of Ti3C2Tx, exposing more hydrophilic functional groups, thereby improving its hydrophilicity.
Comments 2: “The contact angle of Cu/Ti3C2Tx is much smaller than that of Cu/Ti3C2Tx and Cu, which indicates that the Cu/Ti3C2Tx catalyst has strong hydrophilia. In the hydrogen evolution reaction, the stronger the hydrophilia of the catalyst, the shorter the time that the bubbles generated by the reaction stay on the catalyst surface, and the smaller the size of the bubbles leaving the surface of the catalyst, which is more conducive to the improvement of the efficiency of the electrolysis water reaction.” Here again the term “hydrophilia”.
Author reply:
Thanks for your helpful comments. According to your suggestion, this paragraph has been rewritten.
Corresponding revisions:
Page 9 of manuscript, Lines 277-290
The contact angle is a quantitative measure of the wettability of the surface. To further test the hydrophilicity of Ti3C2Tx, Cu and Ti3C2Tx catalysts, the contact angle between the drop and the catalyst surface is measured. In Figure 7(a), the contact angle of Ti3C2Tx is 142.3°. In Figure 7(b), the contact angle of Cu is 33.1°. In Figure 7(c), the contact angle of Ti3C2Tx is 12.7°. The contact angle of Ti3C2Tx is much smaller than that of Ti3C2Tx and Cu, which indicates that the Cu/Ti3C2Tx catalyst has stronger hydrophilicity. In the light of the SEM and, TEM analysis results, Cu nanoparticles are inserted into the interspace between the Ti3C2Tx layers, effectively inhibiting the accumulation of Ti3C2Tx, exposing more hydrophilic functional groups, thereby improving its hydrophilicity. In the hydrogen evolution reaction, the stronger the hydrophilicity of the catalyst, the shorter the time that the bubbles generated by the reaction stay on the catalyst surface, and the smaller the size of the bubbles leaving the surface of the catalyst, which is more conducive to the improvement of the efficiency of the electrolysis water reaction [49].

Reviewer 2 Report
In the current manuscript, the authors reported a Cu electrocatalyst anchored on Ti3C2Tx MXene for HER purposes. Overall, the study is interesting, appropriate experimentation and characterizations were carried out. However, the manuscript needs extensive English language in terms of spelling, sentence construction, and grammatical errors. Several such issues can be found throughout the manuscript. For instance Line 66 – what does “Taman temperature” mean?
Author Response
Point-by-Point Response to Reviewers’ Comments
For your reference, the detailed point-by-point response to comments from referees is presented below. To save your and the reviewers’ time to quickly read what we have done for the revision, in this response letter, the texts for “Author reply” are marked in blue font, and the texts for “Corresponding revisions” are marked in red font. The revised or added texts for changes have been documented in red font in our revised manuscript.
Thank you for your attention in your busy time!
Response to the comments of referee 2
Comments: In the current manuscript, the authors reported a Cu electrocatalyst anchored on Ti3C2Tx MXene for HER purposes. Overall, the study is interesting, appropriate experimentation and characterizations were carried out. However, the manuscript needs extensive English language in terms of spelling, sentence construction, and grammatical errors. Several such issues can be found throughout the manuscript. For instance Line 66–what does “Taman temperature” mean?
Author reply:
We deeply appreciate the reviewer’s agreement on this work. Your concerns have been carefully considered and we have revised our manuscript with great attention.
Comments 1: In the current manuscript, the authors reported a Cu electrocatalyst anchored on Ti3C2Tx MXene for HER purposes. Overall, the study is interesting, appropriate experimentation and characterizations were carried out. However, the manuscript needs extensive English language in terms of spelling, sentence construction, and grammatical errors.
Author reply:
Thank you for your valuable comments. According to your suggestion, the English grammar of the manuscript has been polished by Professor Zeng Fanjian, a visiting scholar in the English Department of Cambridge University and the postgraduate tutor of the Master of Translation in Jiangxi University of Science and Technology. For Professor Zeng's personal website, please refer to https://ffs.jxust.edu.cn/info/1177/3299.htm
Comments 1: Several such issues can be found throughout the manuscript. For instance Line 66 – what does “Taman temperature” mean?
Author reply:
Thanks for your helpful comments. According to your suggestion, this reference is also replaced, which provides information about Taman temperature
Page 13 of manuscript, Lines 392-393
Wang, A.; Zhang, Y.; Fu, P.; Zheng, Q.; Fan, Q.; Wei, P.; Zheng, L. Achieving strong thermal stability in catalytic reforming of methanol over in-situ self-activated nano Cu2O/ZnO catalyst with dual-sites of Cu species. Journal of Environmental Chemical Engineering 2022, 10, 107676, doi:https://doi.org/10.1016/j.jece.2022.107676.

Reviewer 3 Report
Review attached

Author Response
Point-by-Point Response to Reviewers’ Comments
For your reference, the detailed point-by-point response to comments from referees is presented below. To save your and the reviewers’ time to quickly read what we have done for the revision, in this response letter, the texts for “Author reply” are marked in blue font, and the texts for “Corresponding revisions” are marked in red font. The revised or added texts for changes have been documented in red font in our revised manuscript.
Thank you for your attention in your busy time!
Response to the comments of referee 3
Comments: Review attached
Author reply:
We deeply appreciate the reviewer’s agreement on this work. Your concerns have been carefully considered and we have revised our manuscript with great attention.
Comments 1: I suggest to change the Title to: Copper decorated Ti3C2Tx MXene electrocatalyst for hydrogen evolution reaction
Author reply:
Thank you very much for your advice. According to your suggestion, this Title has been rewritten to: Copper decorated Ti3C2Tx MXene electrocatalyst for hydrogen evolution reaction
Page 1 of manuscript, Lines 2-3
Comments 2: References throughout the manuscript should not be written as superscripts, but rather in line with the text, for instance: instead of [1,2], there should be [1,2].
Author reply:
Thanks for your helpful comments. According to your suggestion, References throughout the manuscript have been rewritten.
Comments 3: Line 36: E0 is 1.23 V (vs. SHE) [reference]
Author reply:
Thanks for your helpful comments. According to your suggestion, New references were added.
Corresponding revisions:
Page 12 of manuscript, Lines 360-373
Rodney, J.D.; Deepapriya, S.; Das, S.J.; Robinson, M.C.; Perumal, S.; Katlakunta, S.; Sivakumar, P.; Jung, H.; Raj, C.J. Boosting overall electrochemical water splitting via rare earth doped cupric oxide nanoparticles obtained by co-precipitation technique. Journal of Alloys and Compounds 2022, 921, 165948, doi:https://doi.org/10.1016/j.jallcom.2022.165948.
Weng, S.X.; Chen, X. A hybrid electrolyzer splits water at 0.8V at room temperature. Nano Energy 2016, 19, 138-144, doi:https://doi.org/10.1016/j.nanoen.2015.11.018.
Comments 4: Lines 48, 49: “…precious metals (Pt, Ru, Pd, Ir, etc.) and their alloys.” There are also a number of supported precious metals nanoparticles, which should be mentioned as a way to reduce the amount of precious metals as given in cited refs. [4,5]
Author reply:
Thanks for your helpful comments. According to your suggestion, the references were replaced.
Corresponding revisions:
Page 12 of manuscript, Lines 374-378
Hughes, J.P.; Clipsham, J.; Chavushoglu, H.; Rowley-Neale, S.J.; Banks, C.E. Polymer electrolyte electrolysis: A review of the activity and stability of non-precious metal hydrogen evolution reaction and oxygen evolution reaction catalysts. Renewable and Sustainable Energy Reviews 2021, 139, 110709, doi:https://doi.org/10.1016/j.rser.2021.110709.
Wu, H.; Feng, C.; Zhang, L.; Zhang, J.; Wilkinson, D.P. Non-noble Metal Electrocatalysts for the Hydrogen Evolution Reaction in Water Electrolysis. Electrochemical Energy Reviews 2021, 4, 473-507, doi:10.1007/s41918-020-00086-z.
Comments 5: Lines 61-63: Avoid the data from this ref. This can be used later if needed for comparison with the results from this work.
Author reply:
Thanks for your helpful comments. According to your suggestion, it has been rewritten.
Corresponding revisions:
Page 2 of manuscript, Lines 62-64
Hyunsik et alobtained two-dimensional copper oxide (CuO) nanosheets on stainless steel plates by chemical bath deposition and air annealing. It was found to have high catalytic activity and stability when used in OER reaction [12].
Comments 6: Line 68: Avoid the expression: “In the meantime,…”, rather “Besides, …”
Author reply:
Thanks for your helpful comments. According to your suggestion, it has been rewritten.
Corresponding revisions:
Page 2 of manuscript, Lines 70
Besides,
Comments 7: Line 74: no need to repeat citing ref [14], since it is already given in the previous sentence.
Author reply:
Thanks for your helpful comments. According to your suggestion, it has been replaced.
Comments 8: Line 80: after ref [15], and before the sentence “The surface of Mxenes…”- the definition of Mxene should be given that involves the following: low dimensional transition metal carbide and nitride (MXenes) Mn+1XnTx (n = 1–3), where, M is transition metal, X is carbide, nitride or carbonitride, Tx is the surface termination groups ((–O), (–F), and (–OH)).
Author reply:
Thanks for your helpful comments. According to your suggestion, this paragraph has been rewritten.
Corresponding revisions:
Page 2 of manuscript, Lines 82-84
MXenes are a diverse group of low dimensional transition metal carbide and nitride with general formula Mn+1XnTx (n=1–3), where, M is transition metal, X is carbide, nitride or carbonitride, Tx is the surface termination groups.
Comments 9: Line 90: hydrogen evolution reaction (HER) –write ether “hydrogen evolution reaction” or “HER”. Both are already given before.
Author reply:
Thanks for your helpful comments. According to your suggestion, all such expressions have been revised in the full text.
Corresponding revisions:
Page 2 of manuscript, Lines 95
Then, the electrocatalyst Cu/Ti3C2Tx suitable for HER was prepared by supporting Cu with Cu/Ti3C2Tx.
Comments 10: Lines 92, 93: Write the full names of the techniques used together with their abbreviations in brackets, when they are mentioned here for the first time (Abstract is regarded as independent). For instance: X-Ray Diffraction (XRD), etc.
Author reply:
Thanks for your helpful comments. According to your suggestion, this paragraph has been rewritten.
Corresponding revisions:
Page 3 of manuscript, Lines 97-99
such as Scanning Electron Microscopy (SEM), Transmission Elec-tron Microscopy (TEM), Brunner Emmet Teller (BET), X-Ray Diffraction (XRD), X-ray Photoelectron Spectroscopy (XPS).
Comments 11: Line 98: According to the template for Metals, instead of 2. Experiment Procedure, write 2.
Author reply:
Thanks for your helpful comments. According to your suggestion, the title has been changed.
Corresponding revisions:
Page 3 of manuscript, Lines 104
- Materials
Comments 12: Lines 103-104: “ …titanium aluminum carbide (Ti3C2Tx) powder…” – check the name of the substance and the formula in brackets - something is wrong , maybe (Ti3AlC2)?
Author reply:
Thanks for your helpful comments. According to your suggestion, this paragraph has been rewritten.
Corresponding revisions:
Page 3 of manuscript, Lines 114
Ti3AlC2
Comments 13: Line 115: Rather: “The illustration of the Cu/Ti3C2Tx synthesis is shown in Figure 1. “
Author reply:
Thanks for your helpful comments. According to your suggestion, this paragraph has been rewritten.
Corresponding revisions:
Page 3 of manuscript, Lines 115-126
The illustration of the Cu/Ti3C2Tx synthetic is shown in Figure 1.
Comments 14: Line 126: The preparation of Cu and Ti3C2Tx is similar to the above. – Why this sentence?
Author reply:
Thanks for your helpful comments. According to your suggestion, this paragraph has been rewritten.
Corresponding revisions:
Page 3 of manuscript, Lines 136-138
To expand the comparison scope, the preparation of Cu and Ti3C2Tx is similar to the above. Cu(NO3)2 solution was directly added to NaBH4 to obtain Cu, and Ti3C2Tx solution after ultrasonic dispersion was directly dried to obtain Ti3C2Tx.
Comments 15: Line 130: The crystalline phases of Ti3AlC2Tx ….- should be Ti3AlC2
Author reply:
Thanks for your helpful comments. According to your suggestion, it has been revised in the revised manuscript
Corresponding revisions:
Page 4 of manuscript, Lines 142
Ti3AlC2
Comments 16: Line 141: Remove the part of the sentence: “The electrochemical device is shown in Figure 2,”, together with Figure 2. This is a standard set-up and there is no need to be presented in the paper (good for Master or PhD thesis).
Author reply:
Thanks for your helpful comments. According to your suggestion, this paragraph has been rewritten.
Comments 17: Subsection “Chemicals” is missing – this should involve all chemical used and their producers.
Author reply:
Thanks for your helpful comments. According to your suggestion, this paragraph has been added.
Corresponding revisions:
Page 3 of manuscript, Lines 105-108
- Materials
All reagents were analytic reagent grade and used as received without additional purification. Copper nitrate trihydrate (Cu(NO3)2·3H2O), Titanium aluminium carbide (Ti3AlC2), lithium fluoride (LiF) and Sodium borohydride (NaBH4) was purchased from Aladdin Corporation.
Comments 18: The organization of Section 3. Results and discussion could be improved according to the following suggestion:
Author reply:
Thanks for your helpful comments. According to your suggestion, Results and discussion has been re-organized.
Corresponding revisions:
Page 10-12 of manuscript, Lines 295-347
3.2. HER activity in alkaline electrolytes
The electrocatalytic properties of the synthesized Cu/Ti3C2Tx electrocatalysts were investigated by the CV measurement at different sweep speed. As shown in Figure 8(a), the area of the closed curve increases with increasing scanning speed. To expand the comparison scope, the Cdl of catalyst was defined by the CV of the synthesized Cu, Ti3C2Tx and Cu/Ti3C2Tx. As shown in Figure 8(b), the Cdl of the Cu/Ti3C2Tx is 29.4 mF cm-2. The Cdl of the Ti3C2Tx and Cu are 9.8 and 2.1 mF cm-2, The Cdl of Cu/ Ti3C2Tx is 3 times higher than that of Ti3C2Tx. Besides, the ECSA of catalyst can be obtained from the Cdl value by a formulated methodology [50]. As shown in Figure 8(c), the ECSA of the Cu/Ti3C2Tx is 735 cm-2. The ECSA of the Ti3C2Tx and Cu are 245 and 52.5 cm-2, The ECSA of Cu/ Ti3C2Tx is 3 times higher than that of Ti3C2Tx. The large ECSA value can usually be attributed to the unique structure of Cu/Ti3C2Tx. Ti3C2Tx supports small Cu nanoparticles, which not only effectively prevents Cu agglomeration, but also effectively improves the specific surface area of Cu/Ti3C2Tx catalyst.
Figure 8. HER performance of Ti3C2Tx, Cu and Cu/Ti3C2Tx series catalysts; (a) CV curves of Cu/Ti3C2Tx at the scan rate of 10, 20, 30, 40, 50, 60, 70, 80, 90, 100 mV s-1, (b) comparisons of Cdl, (c) comparisons of ECSA
Further, the HER performance of Ti3C2Tx, Cu and Cu/Ti3C2Tx was investigated in 1 M KOH solution. In Figure 9(a), Ti3C2Tx exhibits the worst HER activity, Cu/Ti3C2Tx exhibits the best HER performance, and catalytic activity exhibits the following trends: Cu/Ti3C2Tx>Cu>Ti3C2Tx. The improvement of the catalytic performance of Cu/Ti3C2Tx can be attributed to the insert of Cu nanoparticles into Ti3C2Tx, which improves the catalytic activity and enhances the conductivity of Ti3C2Tx‐based catalyst [51]. Cu/ Ti3C2Tx shows the lowest overpotential (η10=128 mV), while Cu is 360 mV and Ti3C2Tx is 541 mV. In addition, the overpotential (η10) of Cu/Ti3C2Tx is lower than that of most of the non-precious catalysts for HER listed in Table 1 [52-59].
Moreover, the Tafel slope is an important parameter in evaluating the mechanistic pathway of the hydrogen evolution reaction. Meanwhile, the lower the value of Tafel, the stronger the HER catalytic activity of the catalyst. In Figure 9 (b), the Tafel slope (Cu/ Ti3C2Tx) is 126 mV dec-1, which is lesser than Ti3C2Tx (468 mV dec-1) and Cu (290 mV dec-1), indicating that the dynamics for HER of Cu/Ti3C2Tx is faster [60]. In addition, the value of the Tafel slope infer the rate of controlling step about the reaction, and the Tafel slope of the Cu/Ti3C2Tx catalyst is 126 mV dec-1 corresponding to the Volmer-Heyrovsky mechanism, while the rate controlling step of this reaction is a Volmer step [61,62].
Figure 9. HER performance of Ti3C2Tx, Cu and Cu/Ti3C2Tx series catalysts; (a) LSV curves at the scan rate of 5mV s-1, (b) the corresponding Tafel plots, (c) stability test chart of Cu/Ti3C2Tx, (d) comparisons of mass activity of Cu/Ti3C2Tx.
It is worth noting that the HER reaction under alkaline conditions involves the dissociation of water molecules (Volmer reaction), and water molecules must first adsorb and dissociate on the surface of the catalyst to form adsorption states of Hads and OH¯ [63]. In the light of the SEM, TEM, XRD and XPS analysis results, Cu, CuO and Cu(OH)2 contained in Cu/Ti3C2Tx promote the water dissociation, and improve the overall efficiency of water electrolysis. Figure 9(c) depicts the LSV curve after 1000 CV cycles under alkaline conditions with a current density of 9.4 mA cm-2 at an overpotential of 128 mV. Compared with the current density of Cu/Ti3C2Tx before the test, the current density of Cu/Ti3C2Tx at an overpotential of 128mV after the test only decreases by 0.6 mA cm-2. As shown in Figure 9(d), the mass activity of Cu/Ti3C2Tx at an overpotential of 128mV is 16.95 A/gCu. After the test, the mass activity of Cu/Ti3C2Tx decreases only 1.02 A/gCu, indicating that Cu/ Ti3C2Tx have better stability.
catalyst |
overpotential (η10) |
References |
Cu/Ti3C2Tx |
128 mV |
This work |
Mo-Ti2Cu3 |
133 mV |
[52] |
Cu2(OH)PO4/Co3(PO4)2·8H2O |
138 mV |
[53] |
Cu2O/g-C3N4 |
148.7 mV |
[54] |
PEDOT@Mn-Salen COFEDA |
150 mV |
[55] |
Cu3P |
155 mV |
[56] |
fs-Cu/MoS2 |
181 mV |
[57] |
Cu(OH)2@FCN MOF/CF |
290 mV |
[58] |
Cu2S |
330 mV |
[59] |
Table 1. The overpotential of non-precious metal-based HER catalysts.
Comments 18:
3.1. Characterization of Cu/Ti3C2Tx catalyst
- Before proceeding with the results for each technique used, write a few introducing sentences about
characterization.
3.1.1. SEM
3.1.2. TEM
3.1.3. BET
3.1.4. XRD
3.1.5. XPS
3.1.6. Contact angle test
-Since HER activity in alkaline electrolyte (not plural for electrolyte) belongs to the Section 3. This one
should be:
3.2. HER activity in alkaline electrolyte
Author reply:
Thanks for your helpful comments. According to your suggestion, all of these mentioned statements have been rewritten.
Corresponding revisions:
3.1. Characterization of Cu/Ti3C2Tx catalyst
Page 5 of manuscript, Line 175
3.1.1. SEM
The SEM images clarify the morphology and microstructures of Ti3C2Tx and Cu/Ti3C2Tx.
Page 6 of manuscript, Line 193
3.1.2. TEM
For more detailed structural information of the catalyst, the synthesized Ti3C2Tx and Cu/Ti3C2Tx nanostructures were investigated by TEM.
Page 6 of manuscript, Lines 208-209
3.1.3. BET
The specific surface area of Ti3C2Tx, Cu and Cu/Ti3C2Tx was detected by N2 physical adsorption instrument. N2 adsorption isotherms of Ti3C2Tx, Cu and Cu/Ti3C2Tx was given in Figure 4
Page 7 of manuscript, Lines 226-227
3.1.4. XRD
The crystal structure of the Ti3AlC2, Ti3C2Tx and Cu/Ti3C2Tx can be reflected by the XRD diffraction pattern.
Page 8 of manuscript, Line 254
3.1.5. XPS
The surface chemical composition of the Cu/Ti3C2Tx catalyst was identified by XPS, which can further investigate the oxidation state and composition of Cu/Ti3C2Tx.
Page 9 of manuscript, Line 277
3.1.6. Contact angle test
The contact angle is a quantitative measure of the wettability of the surface.
3.2. HER activity in alkaline electrolyte
Page 10 of manuscript, Lines 295-296
The electrocatalytic properties of the synthesized Cu/Ti3C2Tx electrocatalysts were investigated by the CV measurement at different sweep speed.
Page 11 of manuscript, Lines 312-313
Further, the HER performance of Ti3C2Tx, Cu and Cu/Ti3C2Tx was investigated in 1 M KOH solution.
Page 11 of manuscript, Lines 321-322
Moreover, the Tafel slope is an important parameter in evaluating the mechanistic pathway of the hydrogen evolution reaction.
Comments 19: line 214: peak at 19.3° , but: line 216, it is at 19.1° . Is it OK?
Author reply:
Thanks for your helpful comments. According to your suggestion, this mistake has been corrected.
Corresponding revisions:
Page 7 of manuscript, Lines 230
19.1°
Comments 20: Figure 6 - provide description for (c) in Caption.
Author reply:
Thanks for your helpful comments. According to your suggestion, this mistake has been corrected.
Corresponding revisions:
Page 8 of manuscript, Lines 252
Figure 5. XRD pattern of (a) Ti3AlC2 and Ti3C2Tx, XRD pattern of (b) Ti3C2Tx and (c) Cu/Ti3C2Tx.
Comments 21: 21. HER section is rather confusing with respect to both Figure 9 and the corresponding text.
This part has to be extensively revised. The following suggestions might be helpful:
Figure 9. It consists of too many parts, which do not belong to the same set of results. Therefore, it
should be divided, for instance:
- the first figure: (a) Cyclic voltammetry (now missing) should be presented together with capacity
measurements- now (d), and comparison of the capacity and ECSA- now (e and f). A proper description
should be given.
- the second figure - LSV measurements -now (a) should be presented together with Tafel plots – now
(c), and stability measurements and mass activity- now (h). Scan rate should be given in Captions where
needed (CV, LSV).
Current Figure 9b can be removed, since the values are given in the text, and the difference in the
overpotentials is visually clear from LSV curves. Also, current Figure 9d can be removed, since the values
are given in the text.
Author reply:
Thanks for your helpful comments. According to your suggestion, all of these mentioned statements have been rewritten.
Corresponding revisions:
Page10 of manuscript, Lines 308-311
Figure 8. HER performance of Ti3C2Tx, Cu and Cu/Ti3C2Tx series catalysts; (a) CV curves of Cu/Ti3C2Tx at the scan rate of 10, 20, 30, 40, 50, 60, 70, 80, 90, 100 mV s-1, (b) comparisons of Cdl, (c) comparisons of ECSA
Page11 of manuscript, Lines 329-333
Figure 9. HER performance of Ti3C2Tx, Cu and Cu/Ti3C2Tx series catalysts; (a) LSV curves at the scan rate of 5mV s-1, (b) the corresponding Tafel plots, (c) stability test chart of Cu/Ti3C2Tx, (d) comparisons of mass activity of Cu/Ti3C2Tx.
Comments 22: Table 1. Check the references given in the Table [49-56] with those in the text.
Author reply:
Thanks for your helpful comments. According to your suggestion, references have been rewritten.
Corresponding revisions:
Page 12 of manuscript, Lines 346-347
Table 1. The overpotential of non-precious metal-based HER catalysts.
catalyst |
overpotential (η10) |
References |
Cu/Ti3C2Tx |
128 mV |
This work |
Mo-Ti2Cu3 |
133 mV |
[52] |
Cu2(OH)PO4/Co3(PO4)2·8H2O |
138 mV |
[53] |
Cu2O/g-C3N4 |
148.7 mV |
[54] |
PEDOT@Mn-Salen COFEDA |
150 mV |
[55] |
Cu3P |
155 mV |
[56] |
fs-Cu/MoS2 |
181 mV |
[57] |
Cu(OH)2@FCN MOF/CF |
290 mV |
[58] |
Cu2S |
330 mV |
[59] |

Reviewer 4 Report
1.- It seems that Figures a and b in SEM micrographs were not compared with the same magnification scale; please add the magnification data. In addition, figure b is not good; try to improve the image quality.
2.- Line 189, "Cu0 can exhibit certain activity for water dissociation, especially for small 189 size Cu0 particles". You can not predict the activity for water splitting just based on the particle sizes... please remove this paragraph.
3.- XRD analysis is not useful to corroborate the synthesis of the expected materials. Figure 6b shows that Ti3C2Tx is mainly composed of Ti3AlC2 and TiC; thus, the resulting material can not be described as Ti3C2Tx . Figure 6c shows the main diffraction peaks for Cu species, but there is no analysis for Ti3C2Tx. How can we ensure the material Cu/Ti3C2Tx was synthesized successfully? The figure caption for figure 6 does not mention anything about figure 6c.
4.- Electrochemical tests show bad HER results; the best material presents very high overpotential. It can not be compared to efficient HER electrocatalysts.
5.- The Tafel plots were obtained by plotting the i vs E data in a high overpotential region 0 to -0.4 V vs RHE, this is not correct, Tafel plots must shows the low-overpotential region.
Author Response
Point-by-Point Response to Reviewers’ Comments
For your reference, the detailed point-by-point response to comments from referees is presented below. To save your and the reviewers’ time to quickly read what we have done for the revision, in this response letter, the texts for “Author reply” are marked in blue font, and the texts for “Corresponding revisions” are marked in red font. The revised or added texts for changes have been documented in red font in our revised manuscript.
Thank you for your attention in your busy time!
Response to the comments of referee 4
Comments 1: 1.- It seems that Figures a and b in SEM micrographs were not compared with the same magnification scale; please add the magnification data. In addition, figure b is not good; try to improve the image quality.
Author reply:
I'm very sorry. This is a mistake caused by our negligence. In fact, it was previously overlooked that the magnification of Ti3C2Tx and Cu/Ti3C2Tx is different, resulting in them being incomparable. Since only Cu/Ti3C2Tx has been analyzed by SEM-EDS, we ultimately retain it as a reference.
Corresponding revisions:
Page 5 of manuscript, Lines 175-191
The SEM images clarify the morphology and microstructures of Cu/Ti3C2Tx. Ti3C2Tx exhibits a distinct layered structure [22]. After HF etches, various surface terminations are formed on the surface of Ti3C2Tx [23], which can provide abundant sites for Cu atoms anchored on the Ti3C2Tx. Cu2+ can be reacted with surface terminations of Ti3C2Tx, which transforms into Cu1+ through self-induced redox reactions [24]. Figure 2(a) is the SEM image of Cu/Ti3C2Tx. These particles loaded on the surface of Ti3C2Tx, are Cu, which can be inserted into the gaps between the layers [25]. In addition, Figure 2(b) displays the elemental distribution maps of Cu/Ti3C2Tx, showing the distribution of elements in Cu/Ti3C2Tx by SEM-EDX, and indicating the presence of C, O, Ti, and Cu distributed uniformly throughout the catalyst surface. In addition, the estimated atomic contents of C, O, Ti and Cu are about 26.98%, 33.72%, 9.70% and 29.59%, respectively. The atomic proportion of O is the highest in the Cu/Ti3C2Tx, indicating a potential presence of oxidized copper.
Figure 2. SEM image of (a) Cu/Ti3C2Tx and (b) EDS mapping of Cu/Ti3C2Tx
Comments 2: Line 189, "Cu0 can exhibit certain activity for water dissociation, especially for small size Cu0 particles". You can not predict the activity for water splitting just based on the particle sizes... please remove this paragraph.
Author reply:
Thanks for your helpful comments. According to your suggestion, this paragraph has been removed
Comments 3: XRD analysis is not useful to corroborate the synthesis of the expected materials. Figure 6b shows that Ti3C2Tx is mainly composed of Ti3C2Tx and TiC; thus, the resulting material can not be described as Ti3C2Tx . Figure 6c shows the main diffraction peaks for Cu species, but there is no analysis for Ti3C2Tx. How can we ensure the material Ti3C2Tx was synthesized successfully? The figure caption for figure 6 does not mention anything about figure 6c.
Author reply:
Thanks for your helpful comments. According to your suggestion, the XRD pattern has been redrawn and illustrated in the revised paper.
Corresponding revisions:
Page 7 of manuscript, Lines 231-232
Besides, there appear a new peak at 6.1° indicating that Ti3AlC2 has successfully transi-tioned to Ti3C2Tx [33,34].
Page 7 of manuscript, Lines 246-247
Besides, a peak appears at 60.7° related to Ti3C2Tx [34], which suggests Cu/ Ti3C2Tx has been successfully synthesized. In addition,
Page 8 of manuscript, Lines 250-252
Figure 5. XRD pattern of (a) Ti3AlC2 and Ti3C2Tx, XRD pattern of (b) Ti3C2Tx and (c) Cu/Ti3C2Tx.
Comments 4: Electrochemical tests show bad HER results; the best material presents very high overpotential. It can not be compared to efficient HER electrocatalysts..
Author reply:
Thanks for your helpful comments. According to your suggestion, this title has been rewritten. Copper decorated Ti3C2Tx MXene electrocatalyst for hydrogen evolution reaction. Compared with noble metal catalysts, the HER performance of Cu/Ti3C2Tx is indeed not efficient enough, but a method of non-precious metal catalyst was developed, and we will continue to improve its activity in the future
Corresponding revisions:
Page 1 of manuscript, Lines 2-3
Copper decorated Ti3C2Tx MXene electrocatalyst for hydrogen evolution reaction
Comments 5: The Tafel plots were obtained by plotting the i vs E data in a high overpotential region 0 to -0.4 V vs RHE, this is not correct, Tafel plots must shows the low-overpotential region.
Author reply:
Thanks for your helpful comments. According to your suggestion, this figure has been replaced.
Corresponding revisions:
Page 11 of manuscript, Line 329

Round 2
Reviewer 3 Report
The manuscript is significantly improved and can be accepted for publication in its present form.
Reviewer 4 Report
After modifications, the manuscript can be published in its current form.